# A Review of Coffee By-Products Including Leaf, Flower, Cherry, Husk, Silver Skin, and Spent Grounds as Novel Foods within the European Union

**DOI:** 10.3390/foods9050665

**Published:** 2020-05-21

**Authors:** Tizian Klingel, Jonathan I. Kremer, Vera Gottstein, Tabata Rajcic de Rezende, Steffen Schwarz, Dirk W. Lachenmeier

**Affiliations:** 1Chemisches und Veterinäruntersuchungsamt (CVUA) Karlsruhe, Weissenburger Strasse 3, 76187 Karlsruhe, Germany; tizian.klingel@cvuaka.bwl.de (T.K.); jonathan.kremer@cvuaka.bwl.de (J.I.K.); vera.gottstein@cvuaka.bwl.de (V.G.); tabata.rajcicderezende@cvuaka.bwl.de (T.R.d.R.); 2Coffee Consulate, Hans-Thoma-Strasse 20, 68163 Mannheim, Germany; schwarz@coffee-consulate.com

**Keywords:** coffee by-products, novel food, coffee flower, coffee leaves, coffee pulp, husk, green coffee, coffee silver skin, spent coffee grounds, cascara

## Abstract

The coffee plant *Coffea* spp. offers much more than the well-known drink made from the roasted coffee bean. During its cultivation and production, a wide variety of by-products are accrued, most of which are currently unused, thermally recycled, or used as animal feed. The aim of this review is to provide an overview of novel coffee products in the food sector and their current legal classification in the European Union (EU). For this purpose, we have reviewed the literature on the composition and safety of coffee flowers, leaves, pulp, husk, parchment, green coffee, silver skin, and spent coffee grounds. Some of these products have a history of consumption in Europe (green coffee), while others have already been used as traditional food in non-EU-member countries (coffee leaves, notification currently pending), or an application for authorization as novel food has already been submitted (husks, flour from spent coffee grounds). For the other products, toxicity and/or safety data appear to be lacking, necessitating further studies to fulfill the requirements of novel food applications.

## 1. Introduction

Coffee is one of the most popular beverages and largest traded commodities worldwide [1]. Based on its aromatic flavor and the beneficial effects of caffeine as well as other ingredients, millions of people consume coffee on a daily base. The two main types of coffee, *Coffea arabica* L. and *Coffea canephora* Pierre ex A. Froehner, are grown in tropical and subtropical regions. Brazil, Vietnam, Colombia, and Indonesia are the main coffee-producing countries [2].

The production of coffee starts with the harvest of coffee cherries. After dry or wet processing, the obtained product is green coffee, which is the standardized trading form [3]. However, the industrial production of coffee generates considerable quantities of by-products such as cherry husks, cherry pulps, silver skin, and afterwards, spent coffee [4]. Due to the considerably decreased coffee price, it might be beneficial for coffee farmers and the coffee industry to market these by-products, which were predominantly put to waste in the past.

Currently, these efforts include using these coffee by-products in the food sector. For marketing within the European Union (EU), it is of utmost importance to clarify whether these by-products need to obtain an approval as novel food.

The term “novel food” means food and/or food ingredients which are comparatively new on the European market and therefore have no history of use as safe food for human consumption. To protect human health and consumer interests, the European Novel Food legal framework applies. The Regulation defines “novel food” as any food which has not been used for human consumption to a significant degree within the Union before 15 May 1997 [5].

The legal framework for the definition and authorization of novel foods can be found in regulation (EU) No 2015/2283 and in the two implementing regulations (EU) No. 2017/2468 and No. 2017/2469 [5,6,7]. Article 3(2a) of regulation (EU) No 2015/2283 defines food categories from which novel foods may originate; in the case of coffee by-products, the plant products referred to in subsection (iv) are particularly important. Novel foods that have successfully undergone the authorization procedure within the meaning of Article 10 of regulation (EU) No 2015/2283 are listed in a positive list in the annex to the commission implementing regulation (EU) 2017/2470 [8]. Furthermore, for novel foods belonging to the category of “plants”, there is the possibility to make a simplified notification of a “traditional food from a third country” if the novel food has a “history of safe food use” in a country outside the European Union, according to Articles 14 and 15 of regulation (EU) No 2015/2283. This notification could be quicker and easier than the full application for authorization, but in practice it is often very difficult for the applicant to provide evidence of safe use over 25 years. Both the notification and the application require a description of the novel food and its production, appropriate analysis methods, as well as analytical and toxicological data to demonstrate that there is no safety risk to human health [5].

The European Commission is responsible for authorizing novel foods and, as part of the procedure, can ask the European Food Safety Authority (EFSA) to conduct a scientific risk assessment to establish their safety. EFSA carries out its safety assessment based on dossiers provided by applicants [9].

The Novel Food status of many products is listed in the Novel Food catalogue of the European Commission. The entries are based on current, coordinated information from the EU member states. There is also an entry for the coffee plant (Figure 1).

Obviously, coffee itself (roasted coffee) is not novel due to its history of consumption before 1997; however, all other materials—including green coffee—must be assessed in detail. Therefore, this article will provide a critical review of coffee by-products that may be used as foods, with special reference to their legal status within the EU and potential options for producers to obtain approval according to the Novel Food Regulation.

## 2. Materials and Methods

For this review, electronic searches of the literature were conducted, including the databases PubMed and Google Scholar. A wide range of search terms were used, including coffee, coffee pulp, coffee leaf, green coffee, coffee husk, coffee flowers, coffee by-products, spent coffee grounds, coffee cherry, coffee silver skin. The search process uncovered 149 articles, books, laws, patent specifications, and internet contributions. Only food uses of the coffee by-products are reviewed. Other uses such as industrial uses of coffee wood or coffee twigs for energy were excluded. Green coffee was included, while not strictly being a coffee by-product, because of its potential novel food status.

## 3. Coffee By-Products

The coffee plant is a special crop which, starting in Ethiopia, has now covered large areas of tropical and subtropical cultivation [11]. Coffee belongs to the family *Rubiaceae* and the tribe *Coffeeae* [12]. Currently, more than 100 species of the genus *Coffea* are known, with the two varieties *C. arabica* and *C. canephora* being the most economically exploited species [13]. The evergreen coffee plants produce white flowers which, after fertilization, produce mainly red fruits called coffee cherries [11]. About 3% of *C. arabica* also produce fully ripe yellow cherries. Other coffee species can also produce black or black-and-yellow-striped cherries. Most varieties need 210 days from blossom to ripe cherry (e.g., bourbon or catuai), whereas dwarf varieties (such as caturra and villa sarchi) require some days less, and tall and giant varieties (maragogype and pacamara) around 230–240 days. The structure of a coffee cherry is shown in Figure 2. Under the leathery epicarp (skin), which is green in the beginning and turns red with increasing ripeness, is the soft sweet mesocarp (pulp). Beneath this is the strongly developed endocarp (parchment), enclosing the two seeds (endosperm) called coffee beans, which are covered by their silver skin, a thin tegument [14].

After harvesting the coffee cherries, there are different ways to produce green coffee (Figure 3). In dry processing, the fruits are spread out immediately after harvesting and dried in the sun. Then, the dried pulp, parchment, and parts of the silver skin are removed with peeling machines. This waste is called husk. The much more complex wet method can be differentiated in three different sub-forms: i. pulped natural, ii. semi-washed, and iii. fully washed. In the first step of all wet processing methods, the pulp of the coffee cherries is squeezed off the parchment beans in a pulper. The differentiation between the sub-forms is directly connected to the treatment of the adhering mucilage. In the “pulped natural” method (sometimes also referred to as “semi-dry” method), all mucilage is left on the parchment and dried on it, in order to allow the sugars to enter the coffee bean by osmosis and create a very sweet coffee. In both the “semi-washed” and the “fully washed” method, the mucilage is removed; the semi-washed technique is a mechanical cleaning, i.e., using brushes and the friction of the parchment beans amongst themselves. In the “fully washed” method, the mucilage is broken down by microorganisms, which occur naturally in the environment of the coffee pulping stations and the coffee plants. Bacteria, yeast, and fungi break down the sugars and the pectins within the mucilage and produce acids and alcohols, thus creating a broader spectrum of aromas, while the sweetness is decreased. It is still uncommon to use controlled fermentation as done in the wine, beer, cheese, or bakery industry. In all cases, fermentation is a “spontaneous fermentation”, while the created flavors differ a lot depending on the weather and other influencing factors during harvest and processing.

During roasting of the coffee beans, which usually no longer takes place in the countries of origin, the remaining silver skin still adhering to the beans accrues as waste [15]. Spent coffee grounds are another type of waste produced either by coffee consumers or by the industrial production of instant coffee [11]. As a result, a large quantity of by-products is generated during the production of coffee, and their recycling is becoming more and more important. Parts of the coffee plant such as flowers, leaves, twigs and wood are equally classified as coffee by-products, because they arise during cultivation. The most important by-products for food production are listed in Table 1.

### 3.1. Flowers

Coffee plants form white, multi-flowered, cymose inflorescences. Every year, an adult coffee tree produces 30,000–40,000 flowers [16] (according to the authors’ own observations, this number is probably for *C. arabica*, with *C. canephora* likely expected to have a higher number of flowers). Because of colleters, which produce a glue-like secretion, the flowers protect the plant from dehydration and insects [58]. When the flower is starting to wilt off the plant, the blossoms are plucked, so that the payload of coffee cherries is not affected [59]. After drying, the flowers could be brewed as tisane. Beside high values of total phenolic content, dried flowers obtained from *C. canephora* contain about 1 g caffeine/100 g dry weight and also about 1 g trigonelline/100 g dry weight [16]. Research in the field of coffee flowers is lacking, so there are no toxicologically data available.

### 3.2. Leaves

The shiny, waxed leaves on the side of the main stem of the coffee plant are typically green [60]. For some species or varieties, the young leaves can be bronzed-tipped and are green or dark-green when aging but never brown, except when they are stressed or when they fall down. The leaves of *C. arabica* and *C. canephora* experience a life time of 8 months and 7–10 month, respectively, until they reach a length of up to 15 cm [61] (this literature value of 15 cm is probably for *C. arabica*, while *C. canephora* leaves can reach a length of 30 cm and those of *Coffea liberica* W. Bull ex Hiern even up to 50 cm).

The leaves of the coffee plants have been used for a long time to prepare tea-like drinks. Especially in West Sumatra, Ethiopia, Jamaica, India, Java, and South Sudan, the infusion is consumed as a traditional food [18,62,63]. Regarding species, *C. arabica* leaves have probably been used in Ethiopia, while *C. canephora* leaves have been used in India and West Sumatra. According to Novita et al. the drink is called “kahwa daun” or “kawa” in Sumatra [20]. In Yemen, it is called “giser” and in Ethiopia “kuti”, “jeno”, or “jenuai” [64]. Before preparing the drink, there are different production methods to obtain coffee leaf tea. Most of them include leaf steaming, rolling, and drying [65]. However, some manufacturers work under a protective gas atmosphere to preserve the ingredients from oxidation [66,67,68,69]. Alternatively, the leaves can also be fermented [65]. In some production methods, the drying process is supplemented by a roasting process [70,71]. To produce a drink, the tea has to be extracted with water or aqueous solvents [19]. Chen et al. found that a variation of the production method (white, green, oolong, black; based on the production of tea from *Camellia sinensis* (L.) Kuntze) produced different phytochemical compositions and different flavors [61].

Besides their use as tea, coffee leaves are also used in ethnomedicine in the originating countries. The composition of coffee leaves was recently reviewed by Chen [61]. The leaves contain carbohydrates, protein and amino acids, organic acids as well as a large spectrum of ingredients, such as alkaloids, flavonoids, terpenes, tannins, xanthonoids, phenolic acids, flavonoids, phytosterols, amino acids, and carotenoids, which is held responsible for the diverse potential bioactive effects typically known from in vitro studies only, e.g., antioxidant, anti-inflammatory, antihypertensive, antibacterial, and antifungal activities [18,62,64,72,73,74,75]. *C. arabica* leaves contain 1.8 and 3.2 mg/g fresh weight of caffeine in mature and young leaves, respectively [17]. The content considerably changes during aging [76].

Besides the use as food, coffee leaves are being tested in several other new applications, as vehicle perfume [77], facial cleanser [78], tobacco substitute [79,80], animal feed [81], *Lactobacillus* proliferating agent [82,83], packaging material [84], absorbance pad [85], and deodorizer [86].

For risk assessment, a level of 7 mg/g of caffeine in dried leaves and 9.9–10.9 mg/L of caffeine in an aqueous drink prepared from 20 g/L of these leaves was determined by the EFSA in the context of a novel food notification as traditional food from a third country [87]. As specifications for a safe food, EFSA gives a reference of <100 mg/L for chlorogenic acid, <80 mg/L for caffeine, and <700 mg/L for epigallocatechin gallate. The maximum levels for caffeine and chlorogenic acid are based on the recommendations of the applicant, whereas the maximum level for epigallocatechin gallate is based on the observed safe level according to Hu et al. and the intake of green tea [88,89]. The xanthine mangiferin was only found in the leaves of *C. arabica* (5 mg/kg dry weight) and not in those of *C. canephora* [72]. Signs of exocrine pancreas toxicity of mangiferin in rodents were observed at a dose of 1000 mg/kg bodyweight (bw) [90]. Reddeman et al. found no evidence of genotoxicity for mangiferin in the Ames test, but a clastogenic effect could be detected in the chromosomal aberration test. However, this could not be confirmed in an in vivo mammalian micronucleus test with a dose up to 2000 mg/kg bw/day [91]. Thus, the EFSA considers the presence of mangiferin in coffee leaves to be safe [87].

### 3.3. Stems, Twigs, Wood

Stems, twigs, and wood are by-products without potential food uses; minor amounts are tolerated in coffee trade as “defects” and considered a contamination of green coffee [92]. For example, the German coffee regulation tolerates 2 g/kg of non-coffee compounds in roasted coffee [93]. 

### 3.4. Cherry Pulp

A large part of the coffee cherry is the cherry pulp, which also typically contains the skin. It represents around 29% of the dry matter. Cherry pulp is a by-product of all wet processing methods [11,94]. Cherry pulps contain 4%–12% of protein, 1%–2% of lipids, 6%–10% of minerals, and 45%–89% of total carbohydrates [95]. Phenolic compounds and caffeine (1.3%) are also present in considerable amounts [96]. Possible uses of cherry pulps are jam, juice, concentrate, and jelly. Coffee pulp flour can be used for breads, cookies, muffins, squares, brownies, pastas, and sauces [22,23]. Currently, there are no specific data about the toxicity of cherry pulps in the literature. It can be assumed that bioactive compounds, microbial spoilage, and mycotoxin contamination may pose a safety risk.

### 3.5. Parchment

Coffee parchment is a lignocellulosic material that has been rarely studied and not efficiently utilized [57]. Parchment was suggested as an antifungal additive with potential uses for food preservation [57]. Extracts of parchment have also been suggested as biocomponents with antioxidant activity [24]. Recent literature shows the potential of this by-product as a promising low-calorie functional ingredient for dietary fiber enrichment in foods to regulate blood glucose and reduce the concentration of serum lipids [97].

Assessing the food safety of parchment, Iriondo-DeHond et al. detected the mycotoxin ochratoxin A at 2.7 µg/kg [32]. They also investigated the acute toxicity of raw parchment in rats with a single dose of 2000 mg/kg bw and observed no apparent signs of toxicity, abnormal behavior, or mortality [32].

### 3.6. Cherry Husk

During dry processing, coffee cherries are dried in the sun, and then the beans are mechanically removed from the dried husks. The remaining husks are composed of skin, pulp, and parchment [14]. They contain 8%–11% of protein, 0.5%–3% of lipids, 3%–7% of minerals, and 58%–85% of total carbohydrates [95]. The fiber amount contains 24.5% of cellulose, 29.7% of hemicellulose, and 23.7% of lignin [98]. Caffeine and tannins represent around 1% and 5% of the cherry husk, respectively [95].

The production of a refreshing and stimulating beverage called cascara (from Spanish “cáscara”, meaning husk) has been described. During the preparation, the husks are infused with hot water, and the result is commonly known as coffee cherry tea or cascara [26]. Traditional beverages from coffee cherries have been consumed in Yemen (called Qishr) and Ethiopia (called Hashara) [30,31,99]. Despite the Spanish word being commonly used worldwide for coffee husk or its derived beverage, the authors have not observed any traditional use of cascara in Latin America.

The beverage cascara contains 226 mg/L of caffeine and 283 mg/L of gallic acid equivalents, which represents the total polyphenol content [25]. While studying the in vitro biological activity of aqueous extracts of cherry husks, associations with biomarkers of inflammation, oxidative stress, adipogenesis, and insulin resistance were detected [100,101].

Cherry husks can also be used for the cultivation of microorganisms based on their high fiber content. Using fermentation, the production of citric acid, gibberellic acid, and enzymes has been realized [11,38]. Gouvea et al. claimed that cherry husks are a potential residue-based ethanol production source [102]. Additional uses in foods are in spirits [28,29] and dietary fiber supplements [11,32].

Mycotoxins are mainly produced during the drying process of coffee cherries and could be of toxicological concern. Paterson et al. described that ochratoxin A was formed in coffee cherry pericarp while sun-drying and found its highest contaminations in husks [103]. Iriondo-DeHond et al. also detected quantities of ochratoxin A (4.3 µg/kg) in the insoluble fraction of husks [32]. They also investigated the acute toxicity of raw husk in rats with a single dose of 2000 mg/kg bw and observed no apparent signs of toxicity, abnormal behavior, or mortality [32]. The EFSA determined a tolerable weekly intake for ochratoxin A of 120 ng/kg bw [104]. The preparation of cherry husks aiming to avoid microbial spoilage is very important to achieve low mycotoxin amounts. Nevertheless, the investigation of mycotoxins in dried husks is an essential measure for food safety.

### 3.7. Green Coffee

After coffee cherries have been cleaned of skin, pulp, mucilage, and parchment by dry or wet processing, green coffee beans remain, which are traded on the international market [3,14]. These beans are still mostly covered by their silver skin. In a further, optional polishing step, this silver skin can also be removed [105,106]. Green coffee has a mild, green, bean-like aroma [107]. 

The main constituents of green coffee are insoluble polysaccharides (~50%) such as celluloses and hemicelluloses. In addition to complex carbohydrates, it contains mono- and oligosaccharides, oils and waxes (8%–18%), proteins and amino acids (9%–12%), minerals (3%–5%), and polyphenolic compounds [14,106,108,109,110,111,112]. The most common alkaloid in green coffee is caffeine (1%–4%), whose concentration strongly depends on variety and growing conditions, followed by trigonelline (~0.8%) [113,114,115]. Trigonelline is partially degraded during coffee roasting, forming *N*-methylpyridinium ions by decarboxylation and nicotinic acid by demethylation [116,117,118].

Green coffee is included in the context of coffee by-products as novel food because typically only roasted coffee is consumed by end consumers. Nevertheless, green coffee can be marketed as such (without roasting), and an infusion (from non-selective water extraction) can be prepared from the chopped beans [33,34,35,36,37]. Extracts, customized to the needs of the food industry, can generally also be produced by extraction with hot water [119], alcohol [120], or mixtures thereof [107]. Such selective extracts require a separate assessment as novel food. They are available in capsules as food supplements or are used for the preparation of beverages or chewing gum. Due to the high content of chlorogenic acids, which would be partially lost during roasting, the extracts are sold with reference to their potential, diverse, health-promoting effects, some of which have yet to be proven [35]. Watanabe et al. examined green coffee extract in a placebo-controlled human intervention study and found no serious side effects. The subjects received an extract containing 140 mg of chlorogenic acid per day for 12 weeks [121]. In a rat study of green coffee oil, which is rich in kahweol and cafestol, no signs of toxicity could be found. The acute administration of 2000 mg/kg bw and the subacute administration of up to 75 mg/kg bw exhibited no toxicological effects, but the relative weight of heart and thymus increased without histopathological changes [122].

Caffeine is known to affect the central nervous system and is therefore popular for its stimulating effect. However, overdosing can also have adverse effects, so that, especially when placing food supplements on the market, the recommendation for maximum daily intake of caffeine (400 mg/day) by the EFSA should be observed [123]. 

Especially in the production of extracts, it should be noted that mycotoxins can also be enriched. Vaclavik et al. found up to 136.9 μg/kg of ochratoxin A, 20.2 μg/kg of ochratoxin B, 415.0 μg/kg of fumonisin B1, and 395.0 μg/kg of mycophenolic acid in green coffee extracts [124]. 

### 3.8. Silver Skin

The coffee silver skin forms a thin tegument, which is located directly around the two beans of the coffee cherry. It accumulates in large amounts as a by-product of the roasting process. The coffee silver skin is composed to a large extent of dietary fiber (60%–80%) [44]. Pertinent fiber components are cellulose, hemicellulose, and lignin. The main monosaccharides in coffee silver skin hemicellulose are xylose, galactose, arabinose, and mannose. A relevant amount of the total dietary fiber is soluble, which gives coffee silver skin the potential to be used as raw material in the development of functional foods [43]. There is no significant difference of the dietary fiber content between *C. arabica* and *C. canephora* coffee silver skin [32]. The protein content in coffee silver skin is around 18% [41,43]. The ash value is high, which is attributed to minerals. Other nutrients such as fats and reducing carbohydrates are found in lower concentrations [41]. 

The coffee silver skin contains phenolic compounds and has a high antioxidant activity. Additionally, it contains Maillard reaction products, which are formed during the coffee roasting process [41]. The caffeine content in roasted coffee silver skin ranges between about 0.8 and 1.4 g/100 g [125]. Canephora coffee silver skin has a significantly higher caffeine content than Arabica coffee silver skin. Iriondo-DeHond et al. determined the caffeine content in extracts from *C. arabica* and *C. canephora* silver skin. Coffee silver skin extracts also contain other bioactive compounds such as 5-caffeoylquinic acid, a chlorogenic acid, and flavonoids such as rutin [126]. Thus, coffee silver skin has the potential to be used as a natural and sustainable ingredient in foods [32,126].

Primarily coffee silver skin could be deployed as a source of antioxidant dietary fiber in foods. For example, coffee silver skin extracts may be used as a natural colorant and as a source of dietary fiber in biscuits [45]. Furthermore, coffee silver skin may improve the quality, shelf life, and sensory quantities of barbari bread [44]. There are also applications of coffee silver skin in antioxidant beverages [49]. Due to its roasting-derived flavors, it could be used to impart the smoke aroma into other foods, e.g., as a smoked salt (authors’ own observation).

In addition to the abovementioned amounts of caffeine, coffee silver skin also contains acrylamide, as this contaminant is produced during roasting [32,45]. Contamination with mycotoxins, especially ochratoxin A (18.7–34.4 μg/kg) is also possible [127]. Iriondo-DeHond et al. investigated the acute toxicity of an aqueous extract of coffee silver skin in rats with a single dose of 2000 mg/kg bw and observed no apparent signs of toxicity, abnormal behavior, or mortality [32]. The extract also showed no cytotoxicity in the MTT assay on HepG2 cells. Also in the MTS and lactate dehydrogenase (LDH) assays in HaCaT and HFF-1 cells, no in vitro cytotoxic effects could be detected [128]. The comet assay for in vitro genotoxicity testing in HepG2 cells showed that at concentrations up to 1000 µg/mL of coffee silver skin extract, no relevant strand breaks or oxidized bases were induced [126]. In a subacute toxicity study, in which rats received 1 g/kg bw of aqueous coffee silver skin extract orally, no negative effects on hormone secretion and antioxidant or anti-inflammatory biomarkers were observed [129]. 

### 3.9. Spent Coffee Grounds

Spent coffee grounds are formed both during the extraction of coffee powder with hot water to produce a coffee beverage and during the production of instant coffee preparations [11]. For each kg of instant coffee, 2 kg of wet spent coffee grounds waste are produced, which corresponds to an annual amount of about 6,000,000 t worldwide [11,130].

In the food industry, coffee grounds could be used as a source of dietary fiber [54,55] or in bakery products [51,52,53], as well as to produce alcoholic distillates [50]. After drying the spent coffee grounds, subsequent extraction of coffee oil with supercritical CO_2_, and sterilization, a “coffee flour” is produced, which is highly fibrous, high in protein, and gluten-free. It is intended for use as a novel food ingredient in savory and sweet recipes, bakery products, confectionery, snacks, and ready-to-eat products [56]. The novel food approval is currently pending. 

Polysaccharides are the main component of spent coffee grounds, with hemicelluloses accounting for approximately 39 g/100 g dry weight and celluloses for 12 g/100 g dry weight [43,131,132]. Besides sugars, proteins, and minerals, spent coffee grounds also contain fat, which can be extracted as coffee oil and contains the diterpenes kahweol and cafestol [133,134]. Due to the good water solubility of caffeine, lower caffeine contents are measured in spent coffee grounds (0.007%–0.5%) than in roasted coffee. The caffeine content depends on coffee variety and extraction process [135,136]. 

The two food processing contaminants acrylamide and hydroxymethylfurfural result from the Maillard reaction during coffee roasting [137]. Spent coffee grounds contain 37.2 ± 0.4 μg/kg of acrylamide and 61.3 ± 0.4 mg/kg of hydroxymethylfurfural (dry weight), respectively [52]. Of alkyl pyrazines, which are formed during roasting by the Maillard reaction, about 70%–80% pass into the domestically prepared coffee beverage [138]. It can therefore be deduced that the rest remains in spent coffee grounds. In humans, alkyl pyrazines are metabolized to their corresponding carboxylic acids and almost completely excreted renally [139]. Spent coffee grounds are susceptible to contamination with mycotoxins, therefore Iriondo-DeHond et al. investigated this. They could not detect aflatoxin B1 and enniatin B but found ochratoxin A in an amount of 2.31 µg/kg. They also carried out an acute toxicity study in rats, in which no visible signs of toxic effects were observed at a dose of 2000 mg/kg bw of spent coffee grounds [134].

## 4. Novel Food Status of Coffee By-Products

### 4.1. Evidence for Human Consumption of Coffee By-Products within the EU

In general, food can be placed on the European Union (EU) market without prior authorization. An exception is made for novel foods which were not used for human consumption to a significant degree within the EU before 15 May 1997 [5]. Green unroasted coffee beans, as well as “white coffee” made from them by a non-selective water extraction, were on the market as a food or food ingredient and consumed to a significant degree before 15 May 1997 [10]. Thus, their access to the EU-market is not subject to the novel food regulation (except for certain extracts, see Section 3.7).

Some of the other coffee by-products may also have been used to some degree within the EU according to historical sources. Coffee leaf tea may have been consumed in England in the 19th century. According to a 1864 handbook on coffee planting in southern India, “coffee leaf tea has already been introduced into England” [140], while the *Experiences of a Planter in the Jungles of Mysore* (1871) reported that “roasted coffee leaves have been sold in London as ordinary tea” and “a London broker reported on some Ceylon coffee-leaf tea” [141].

According to McCabe, the French form of “à la sultane coffee” developed by Nicolas de Bois-Regard Andry (1658–1742), a medical doctor and professor at the Collège du Roi, was an infusion of coffee husks [142]. In his German textbook on coffee and other stimulants, Neumann (1735) also mentions the preparation of a “Café à la Sultane” or Sultanin Café, a drink made from roasted coffee husks [143]. A more detailed description of the preparation of a “Café à la Sultane” is given by the “Naturforschende Gesellschaft zu Danzig” (Danzig Research Society) (1756). After removing the seeds, the pulp of the coffee cherry is dried in a pan (but not roasted), then hot water is added and boiled. The resulting beverage resembles a tea in its color [144]. 

However, these historical books do not provide compelling evidence that the food was used for human consumption to a significant degree within the Union before 15 May 1997. The EU’s guidance document provides assistance on the question of what can be defined as “human consumption to a significant degree” [145]. The coffee industry was obviously not able to provide the required evidence (such as trade or import documents), so that novel food approval appears necessary for coffee leaves and husks, despite this evidence from England, France, Germany, and Poland. The EU novel food catalogue therefore classifies the tea use of dried berries of *Coffea* spp. (coffee cherry tea) as novel [10]. However, other coffee by-products are currently not mentioned in the EU novel food catalogue (Figure 1), although for some by-products applications for authorization have been submitted, or notifications as traditional foods from third countries have been made (coffee leaves, husks, flour from spent coffee grounds). The catalogue is a non-exhaustive, non-legally binding list, based on up-to-date information on the market situation, which is intended only as a guide for marketers and authorities. Approved novel foods are included in the union list [8].

The authors own judgements about the current novel food statuses of coffee by-products are listed in Table 2.

### 4.2. The Way Forward: Novel Food Approval of Coffee By-Products

For some of the coffee by-products, especially the leaves, traditional food usage in third countries has been known. People in Ethiopia, South Sudan, Liberia, Indonesia, and Jamaica traditionally consume herbal infusions made from coffee leaves. AM Breweries IVS already made a notification for coffee leaves according to Articles 14 and 15 of regulation (EU) No 2015/2283 as a traditional food from a third country [87,146]. A decision regarding this notification is expected in 2020 by the European Commission, so that infusions from coffee leaves are soon to be expected as being legally marketable within the EU as food.

For coffee flour, made from defatted spent coffee grounds, and coffee husk, novel food applications were made under Article 10 of regulation (EU) No 2015/2283. Hence, permission for the use of husk in the EU for the production of non-alcoholic, water-based beverages is currently being examined [147]. The request for coffee flour does not relate to the use of spent coffee grounds as such but refers to a further processed (dried, defatted, sterilized) product made from them and intended for use in savory and sweet recipes, in bakery, confectionary, snacks, and ready-to-eat products [56]. According to the web-based list of applications currently being processed by the European Commission, there is no application pending for spent coffee grounds themselves [148].

Pulp and flowers of *Coffea* spp. are most likely to be classified as novel foods, which are not approved for marketing as food in the EU, and for which no application for authorization has yet been submitted. Whether silver skin is to be judged as novel is not entirely clear, because parts of the silver skin remain in the final product during roasting of the coffee beans and have thus been consumed along with coffee for a long time. It is therefore recommended to consult the authorities to clarify the status of the silver skin (Article 4 (2) of regulation (EU) No 2015/2283) [5]. To this end, the food business operator shall submit a request for consultation to the recipient member state, including a cover letter, technical dossier, supporting documentation, and an explanatory note clarifying the purpose and relevance of the submitted documentation, in accordance with Article 4(1) of implementing regulation (EU) No. 2018/456. If the information provided by the submitting party is complete, the member state shall verify the validity of the consultation in accordance with Article 5. After the validity has been confirmed, the member state decides within four months, possibly with the participation of other member states, on the status of the food according to Art. 6. The commission publishes the decision on its website in accordance with Article 7(2) [149].

## 5. Conclusions

The modern, ecologically oriented society attaches great importance to waste reduction, so it makes sense not to dispose of the by-products of coffee production and to bring them into the value chain [4], if it is proven that this does not pose a safety risk to human health. An added value of the coffee plant could increase social and economic prosperity in poorer coffee-growing regions and work against the decreasing coffee price, which is especially worthwhile in the current times of a global economic crisis. It is estimated that at least 70% of the world’s coffee farmers are no longer able to live sustainably from coffee cultivation because the stock market prices of green coffee are at an all-time low, despite massive crop failures and steadily rising consumption figures. In the coffee market, supply and demand models are overridden by the problems of cash flow during the harvest, which, depending on the degree of mechanization, usually accounts for around 70% of the total production costs of coffee. Since most coffee pickers work as day laborers, it is essential for farmers to have money to pay their workers—a reference to higher prices on the stock exchange is not enough to feed the pickers’ families.

The use of coffee by-products may solve several problems of the coffee farmers. They can offer work over longer periods and at different times of the year, so that ideally the workers can be employed on the farm all year round. Thus, at any time of the year and not only during the harvesting period, which usually lasts 3–5 months, workers would be available to carry out pruning, plant protection, planting, pest monitoring and control, and many other activities. The families would become settled and could set up a better equipped house instead of substandard houses at several different locations. This would create a greater bond between the workers and the farm, with a better understanding of the entire cycle and sustainable management, and allow the workers to live in and with the coffee plantation (also see [18] for example). In the current COVID-19 lockdown, such a model has been specifically advantageous for coffee farms with own workers, while for other farms the access to day laborers has been difficult if not even impossible.

For consumers, the coffee by-products such as tea from cascara or from the leaves offer novel and refreshing beverages that might possibly replicate the observation from 1756 that they “clear the head excellently and lighten the whole body” [144]. The authors also suggest the preparation of coffee leaf tea with a nitro cold brew dispensing equipment as a specifically refreshing beverage for the summer.

## Figures and Tables

**Figure 1 foods-09-00665-f001:**
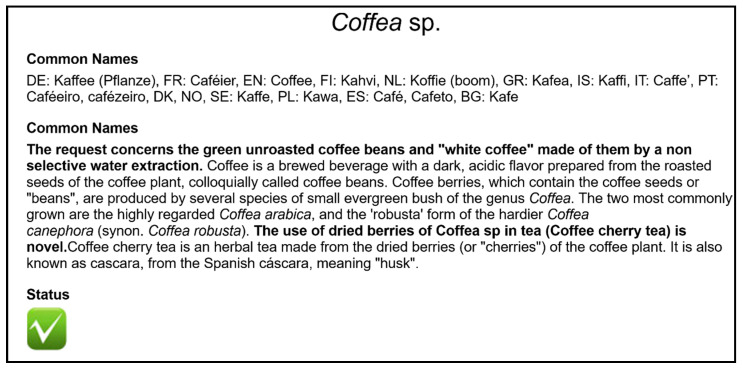
European Union (EU) Novel Food Catalogue entry for *Coffea* (accessed on 31 March 2020) [10].

**Figure 2 foods-09-00665-f002:**
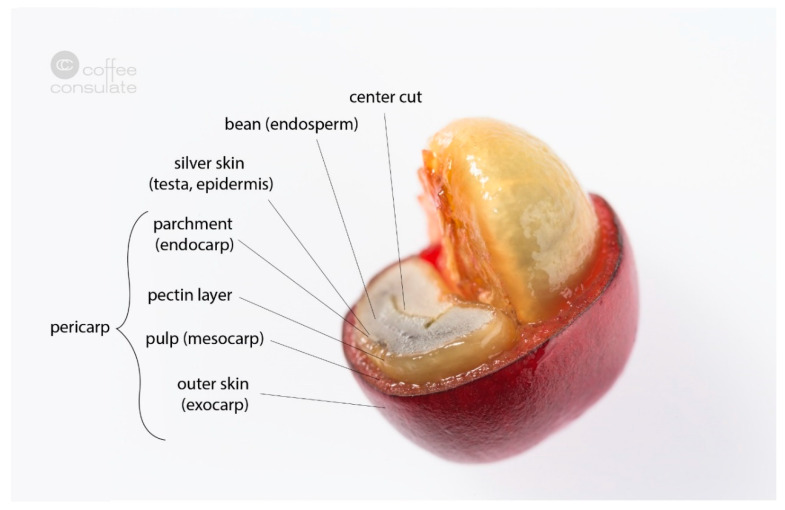
Cross section of a coffee cherry with its different layers.

**Figure 3 foods-09-00665-f003:**
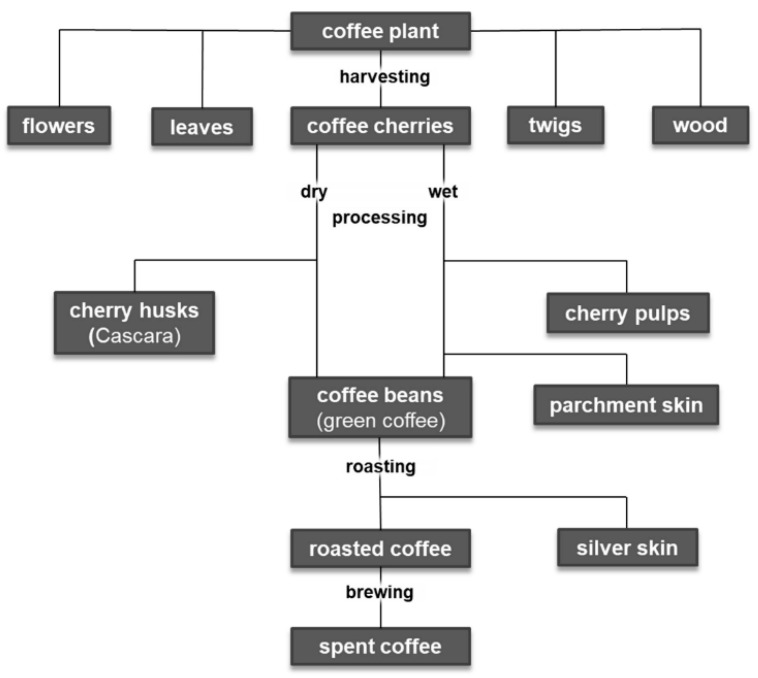
Coffee production-related by-products.

**Table 1 foods-09-00665-t001:** Main uses of coffee by-products in the food sector.

Coffee By-Product		Possible Use as Food
Flowers	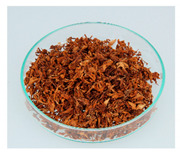	Beverages (tea): “coffee blossom tea” [16]
Leaves	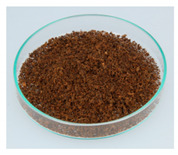	Beverages (tea): “coffee leaf tea” [17,18,19,20,21]
Coffee pulp	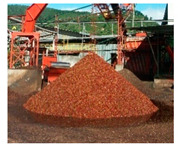	Jam, juice, concentrate, jelly [22]; Coffee pulp flour for breads, cookies, muffins, squares, brownies, pastas, sauces and beverages [23]; spirits/ethanol [24]
Husks, cascara, dried coffee cherries	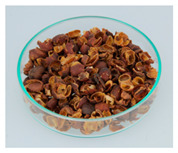	Beverages (tea) [25,26,27]; spirits [28,29]; qishr (mixture with spices) [30,31]; dietary fiber source [11,32]; extraction of caffeine [24]
Green unroasted beans	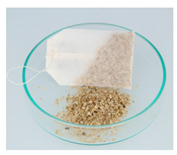	Dietary supplement [33,34,35]; beverages (tea): “white coffee” [36,37]
Silver skin	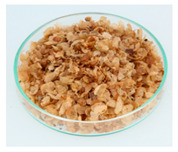	Dietary fiber source [38,39,40,41,42,43]; bakery products (breads, biscuits) [44,45,46,47]; beverages (tea) [48,49]; smoke flavor additive
Spent coffee grounds	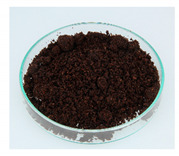	Historical: adulteration of coffee; spirits [50]; bakery products [51,52,53]; dietary fiber source [54,55]
		Defatted *Coffea arabica* seed powder (coffee flour) for savory and sweet recipes, in bakery, confectionary, snacks, ready-to-eat products [56]
Parchment	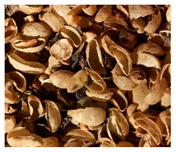	Food preservative, antioxidant [24,57]

**Table 2 foods-09-00665-t002:** Coffee by-products and assessment of their novel food status considering regulation (EU) No 2015/2283.

Coffee By-Product	Novel Food Status ^a^
Flowers	Novel. Probably not a traditional food from third country. Needs approval procedure.
Leaves	Novel, notification for infusion from coffee leaves as traditional food from third country (Ethiopia) submitted by AM Breweries IVS, Amager, Denmark [146].
Coffee pulp	Novel, currently not approved. No application pending. Needs approval procedure.
Husks, cascara, dried coffee cherries	Novel, application submitted by Panama Varietals GmbH, Marchtrenk, Austria [147].
Green unroasted beans	Not novel [10]. The classification also applies to the non-selective water extraction made of them. Selective extracts could be novel.
Silver skin	Unclear. Consultation procedure suggested.
Used coffee grounds	Novel, currently not approved. No application pending. Needs approval procedure.
	Novel, application submitted for a certain “coffee flour” by Kaffee Bueno ApS, Copenhagen, Denmark [56].
Stems, twigs, wood	Non-food material, contamination up to certain levels typically tolerated in the trade of green coffee.
Parchment	Novel, currently not approved. No application pending. Needs approval procedure.

^a^ Authors’ judgement considering the EU Novel Food Catalogue and pending applications/notifications.

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
