# Peer review of "A Review of Coffee By-Products Including Leaf, Flower, Cherry, Husk, Silver Skin, and Spent Grounds as Novel Foods within the European Union"

_foods, 2020, doi:10.3390/foods9050665_

Round 1

Reviewer 1 Report

The manuscript “A review of coffee by-products including leaf, flower, cherry, husk, silver skin and spent grounds as novel foods within the European Union” by Klingel et al., provides an overview of novel coffee products in the food sector and their current legal classification in the European Union (EU). They have reviewed the literature on composition and safety of coffee products and by-products. However, lack of recent information in this field is observed and need to be included.

The paper presents important flaws that reduces its impact and consistency. The authors should make great changes. The comments from the reviewer are listed below:

The section of Materials and Methods should be deleted in this type of article (review). The information given is not relevant by scientific community. On the contrary, the introduction should be lengthened. The reviewer encourages the authors to emphasize the EU framework and EFSA’s role in the “novel foods” in order to become the introduction more consistent.

Figure 2. Please delete cascara (Spanish name) from cherry husk box.

Because one of the main points of this review is the composition and safety of coffee flowers, leaves, pulp, husk, parchment skin, green coffee, silver skin and spent coffee grounds, the authors should be sure include accurate information. For this purpose, the reviewer make the following suggestions:

In line 134, the authors may improve the paragraph adding the phytochemical composition of tea coffee and compare to coffee beverage. The section related to leave composition should be improved.

Line 163 Cherry pulp contains also skin, it should be clarified in the text.

Lines 171-178. In the sub-section of parchment skin, recent literature shows this by-product as potential to be used as a promising low-calorie functional ingredient for dietary fiber enrichment in foods for regulating blood glucose and reducing the concentration of serum lipids. The reviewer suggests to include this information (Benitez et al., Food Res Int., 2019). The authors should complete the composition and the potential physiological properties of this coffee by-product to make consistent this paragraph.

Line 184 tannis in not capital

Line 185 “cáscara” is the write word, please change it through the whole text.

Line 189 “mL” instead of ml, please revise through whole text.

The section of cherry husk should be improved in order to include other findings related to the in vitro biological activity of this aqueous extract into their associations with biomarkers of inflammation, oxidative stress, adipogenesis, and insulin resistance (Rebollo et al., 2019, Antioxidants, Rebollo et al., 2019 Food and Chemical Toxicology)

The sections related to Green coffee and Silver skin need to improve the safe aspects. A more comprehensive discussion is necessary including more studies related to the safety of this product should and their beneficial properties, which are, scarcely discussed.

The section 4. Novel food status of coffee by-products is not well driven, scientifically it is very inconsistent needs a deep improvement to be published in this Journal.

Figure 3 should be omitted and instead, it is already include the web link.

CONCLUSIONS

The authors might entirely rewrite this section. They may write a brief summary at the beginning to focus the readers, but avoiding the use of references.

FINAL COMMENT

As it is,  the authors should rewrite some parts of the manuscript.

Author Response

The manuscript “A review of coffee by-products including leaf, flower, cherry, husk, silver skin and spent grounds as novel foods within the European Union” by Klingel et al., provides an overview of novel coffee products in the food sector and their current legal classification in the European Union (EU). They have reviewed the literature on composition and safety of coffee products and by-products. However, lack of recent information in this field is observed and need to be included.

Thank you for the review. More recent information and references were included in the text as requested (9 new references).

The paper presents important flaws that reduces its impact and consistency. The authors should make great changes.

Thank you for your insight. We have carefully corrected the whole text to correct the specified points for consistency.

The comments from the reviewer are listed below:

The section of Materials and Methods should be deleted in this type of article (review). The information given is not relevant by scientific community.

We disagree that the methods section should be deleted. It is important for the reader to know what was the search strategy and what were the inclusion and exclusion criteria for the review.

On the contrary, the introduction should be lengthened. The reviewer encourages the authors to emphasize the EU framework and EFSA’s role in the “novel foods” in order to become the introduction more consistent.

As requested, some aspects about EU novel food framework and EFSA´s role were included in the introduction.

Figure 2. Please delete cascara (Spanish name) from cherry husk box.

We disagree. The term cascara is much more well known than the Engish term. Most products (e.g. check Amazon) are also traded under the disgnation cascara.

Because one of the main points of this review is the composition and safety of coffee flowers, leaves, pulp, husk, parchment skin, green coffee, silver skin and spent coffee grounds, the authors should be sure include accurate information. For this purpose, the reviewer make the following suggestions:

In line 134, the authors may improve the paragraph adding the phytochemical composition of tea coffee and compare to coffee beverage. The section related to leave composition should be improved.

The section around line 134 was revised. To comprehensively review the composition of coffee leaves, another paper would be necessary (see the excellent review by Chen (2018)). We have pointed out the reference to this review more prominently.

Line 163 Cherry pulp contains also skin, it should be clarified in the text.

Thank you. This information was added to line 163.

Lines 171-178. In the sub-section of parchment skin, recent literature shows this by-product as potential to be used as a promising low-calorie functional ingredient for dietary fiber enrichment in foods for regulating blood glucose and reducing the concentration of serum lipids. The reviewer suggests to include this information (Benitez et al., Food Res Int., 2019). The authors should complete the composition and the potential physiological properties of this coffee by-product to make consistent this paragraph.

Thank you for pointing out this additional reference about the rarely studied parchment. We have included the information as requested.

Line 184 tannis in not capital

Tannins was changed as requested.

Line 185 “cáscara” is the write word, please change it through the whole text.

The Spanish word “cáscara” has been anglicised to cascara, which is commonly used in English scientific texts. This was clarified in line 185.

Line 189 “mL” instead of ml, please revise through whole text.

The symbol for “liter” was corrected throughout.

The section of cherry husk should be improved in order to include other findings related to the in vitro biological activity of this aqueous extract into their associations with biomarkers of inflammation, oxidative stress, adipogenesis, and insulin resistance (Rebollo et al., 2019, Antioxidants, Rebollo et al., 2019 Food and Chemical Toxicology)

Thank you for these new references. The information was included in the text.

The sections related to Green coffee and Silver skin need to improve the safe aspects. A more comprehensive discussion is necessary including more studies related to the safety of this product should and their beneficial properties, which are, scarcely discussed.

We re-searched the literature for data on safety of green coffee and silver skin, but were unable to find more references than already included.

The section 4. Novel food status of coffee by-products is not well driven, scientifically it is very inconsistent needs a deep improvement to be published in this Journal.

We have revised section 4 as requested, see also the requests from reviewer #2. Specifically, we have moved the general parts about introducing the legal framework into the introduction, so that sections 4 only retains the specific parts about coffee by-product regulation. Hopefully, this will have improved the logic and the flow of the paper.

Figure 3 should be omitted and instead, it is already include the web link.

We disagree. This is important information and not many people will click a link while reading a paper. The figure is also not large and the journal does not have length restrictions.

Nevertheless, we decided to move the figure into the introduction, where it might be better fitting, also to improve the flow of section 4 (see previous remark).

CONCLUSIONS

The authors might entirely rewrite this section. They may write a brief summary at the beginning to focus the readers, but avoiding the use of references.

Actually, in looking back at our text, all authors are in agreement that we immensely like our conclusion. As the conclusion section should be able to transport some personal remarks and future outlook, we hope that the reviewer will accept that we have retained the text of the conclusion. Furthermore, we would also like to retain the references, e.g. the anecdotal quote from 1756 about the refreshing effects of coffee by-products.

FINAL COMMENT

As it is, the authors should rewrite some parts of the manuscript.

Thank you for the detailed comments. We hope that our revisions now make the manuscript acceptable for publication.

Reviewer 2 Report

This work gives a completely new view on the exploitation of coffee trees for a purpose other than that of obtaining THE coffee-drink. It presents an excellent review of the different parts of the coffee tree that have been or could be exploited to diversify the interest shown in these trees.

However, the bibliography is sometimes incomplete or misdirected. On the other hand, it would be important that when the contents of compounds and the presence of the compounds are mentioned, the species or variety of coffee tree was mentioned. Indeed, there is a great diversity in the content and the nature of the compounds present according to the variety and the species, as well as according to the environmental conditions of growth and the age of the plant. These details must be included in the new document before the paper can be accepted. The corrections and remarks are carried in the attached pdf document.

Author Response

This work gives a completely new view on the exploitation of coffee trees for a purpose other than that of obtaining THE coffee-drink. It presents an excellent review of the different parts of the coffee tree that have been or could be exploited to diversify the interest shown in these trees.

Thank you.

However, the bibliography is sometimes incomplete or misdirected. On the other hand, it would be important that when the contents of compounds and the presence of the compounds are mentioned, the species or variety of coffee tree was mentioned. Indeed, there is a great diversity in the content and the nature of the compounds present according to the variety and the species, as well as according to the environmental conditions of growth and the age of the plant. These details must be included in the new document before the paper can be accepted. The corrections and remarks are carried in the attached pdf document.

Thank you for the detailed remarks and comments. We have carefully revised the text. See comments below.

Responses to comments/remarks in PDF document:

Page: 2

Line 66: Is it the good reference???

Yes, reference was re-checked.

Line 67: is it the good reference?

Yes, reference was re-checked.

Line 68: A reference is needed: don't forget the work of A. Davis!

A reference to the coffee taxonomy review of A. Davis was included.

Page: 3

Figure 1: The arrow for the mesocarp is not well positioned

We tried to improve the position of the arrow.

Page: 4

Table 1: leaf

Spelling of leaf was corrected throughout.

Page: 5

Line 111: for what species? Is it the same for all the coffee species?

The original reference does not enlighten us about these questions, sorry. According to our own knowledge (no reference found in literature), C. canephora is expected to have a higher number of flowers than C. arabica. We have added this remark into a bracket in line 111.

Line 119: for some species or varieties, the young leaves can be red-brown, and they are green or dark-green when aging, but never brown, except when they are stressed or when they fall down)

Thank you for your insight. We have revised line 119 accordingly.

Line 120: It is not the development that takes this tie, but the life (from the birth to the falling)

Development was changed to life time.

Line 122: the final length of the leaf depends on the species. Is it for C. arabica?

We have added more details about length.

Line 123: Precise what species

Done.

Line 126: Give the number of the reference

All claims can be found in reference #84 at the end of the line.

Line 127: leaf

Spelling of leaf was corrected throughout.

Line 140: This is only true for C. arabica and for mature leaves. The content changes during aging for coffee leaves. See Ashihara et al, Plant Physiol, 1996

The information was specified and the Ashihara reference included. Thank you.

Page: 6

Line 171: Is it "parchment and silver skin" or "silver skin"?

This section is about parchment not silver skin. Skin was deleted.

Line 184: tannins

Done.

Line 186: do you mean: and the result is commonly....

Yes, the line was revised as suggested.

Page: 7

Line 200: body weight (bw)

The abbreviation bw was defined at first use

Line 202 replace by "bw"

Done

Line 212: don't forget the phenolic compounds!! See Anthony et al, 1993, Ky et al, 2001, Campa et al, 2005... and others)

Polyphenols and the references were included into the text. Thank you.

Line 214: A part of the

The sentence was changed.

Line 224: Why don't you mention it before?

Chlorogenic acids are marketed in the context of green coffee for a multitude of claimed effects including weight loss. For this reason, we mention chlorogenic acids in this context first.

Line 224: only 30% is lost!!!

The sentence was changed.

Line 226-238: This part has to be rewritten: what is the relationship between chlorogenic acids, caffeine, diterpenes and toxicity???

The section was revised to make the flow more logical.

Page: 8

Line 252: Before or after roasting

While the reference is not clear about this information, it can be deduced that the analyzed silver skin was roasted as it was purchased from a roastery in Italy. Most if not all commercially available silver skin should be roasted.

Line 256: , a chlorogenic acid

Done.

Line 257: to be?

"To be” was added.

Line 259: what compound gives the colour?

Probably the brown Maillard products formed during roasting. However, the original reference cited contains no information about the nature of the colour.

Line 272-274: this part needs to be mentioned at the beginning of the paragraph, line264.

The part was moved to the position as requested.

Page: 9

Line 313: leaf

Spelling of leaf was corrected throughout.

Line 314: leaf

Spelling of leaf was corrected throughout.

Page: 12

Line 411: Why this quantity?

Typically, the workers have two houses. One at “home” on one on the “farm”. We have revised the sentence to improve clarity.

Line 413: An example is given in the reference 12

Thanks. We have added the reference.

Line 416: leaf

Spelling of leaf was corrected throughout.

Round 2

Reviewer 1 Report

The manuscript “A review of coffee by-products including leaf, flower, cherry, husk, silver skin and spent grounds as novel foods within the European Union” by Klingel et al., needs minor revision.

The reviewer insists on the information of the section of Materials and Methods should be summarized and included in Introduction section.

As suggested in the previous revision, no references are included in the conclusions section, then, the reviewer suggest to change the name of this section to Final Remarks.

Author Response

The manuscript “A review of coffee by-products including leaf, flower, cherry, husk, silver skin and spent grounds as novel foods within the European Union” by Klingel et al., needs minor revision.

The reviewer insists on the information of the section of Materials and Methods should be summarized and included in Introduction section.

According to the authors’ information and the paper template, the materials and methods section should be separate and called exactly that. We not necessarily disagree with this request, but would this leave to editorial discretion or copy-editing changes.

As suggested in the previous revision, no references are included in the conclusions section, then, the reviewer suggest to change the name of this section to Final Remarks.

According to the authors’ information and the paper template, the last section is called “conclusion” and not “final remarks”. According to this “This section is not mandatory, but can be added to the manuscript if the discussion is unusually long or complex.” There is nothing specified about references. We still believe that our text is fitting as a conclusion section. Therefore, we would leave this change to editorial discretion or copy-editing changes.

Reviewer 2 Report

All the modifications have been done. Only two little errors remain in the text. They have been underlined in the attached document.

Author Response

Line 171: Leaves contain

Done

Line 177: change by: between 1.8 and 3.2 mg/g fresh weight in mature and young leaves, respectively.

Done